# Actinic Keratoses: A Prospective Pilot Study on a Novel Formulation of 4% 5-Fluorouracil Cream and a Review of Other Current Topical Treatment Options

**DOI:** 10.3390/cancers15112956

**Published:** 2023-05-28

**Authors:** Ludovica Toffoli, Caterina Dianzani, Serena Bonin, Claudio Guarneri, Fabrizio Guarneri, Roberta Giuffrida, Iris Zalaudek, Claudio Conforti

**Affiliations:** 1Dermatology Clinic of Trieste, Maggiore Hospital, University of Trieste, 34125 Trieste, Italy; 2Department of Plastic Surgery Unit, Section of Dermatology, University Campus Biomedico of Rome, 00128 Rome, Italy; 3Department of Medical Science, Cattinara Hospital, University of Trieste, 34149 Trieste, Italy; 4Department of Biomedical and Dental Sciences and Morpho Functional Imaging, University of Messina, 98124 Messina, Italy; 5Department of Clinical and Experimental Medicine, Dermatology, University of Messina, 98124 Messina, Italy

**Keywords:** actinic keratosis, skin cancer, non-melanoma skin cancer, therapy, 5-Fluorouracil, dermoscopy

## Abstract

**Simple Summary:**

Actinic keratosis (AK) is a premalignant skin lesion caused by chronic sun damage, which usually arises in older people on chronically sun-exposed areas of the body, such as the face, scalp, ears, neck, back of the forearms and hands. Actinic keratosis rarely occurs as a single lesion; most often it can involve multiple lesions in a field of cancerization. Since the AKs may progress to invasive squamous cell carcinoma, treatment of any AK regardless of clinical severity is strongly recommended. The goals of AK therapy are to eradicate clinical and subclinical lesions, prevent the progression into invasive squamous cell carcinoma, provide a good cosmetic outcome and reduce the risk of relapses. Multiple treatment options are available for this condition; however, no gold standard therapy has been established. A novel formulation of 4% 5-Fluorouracil with once daily application proved to be a highly effective and safe treatment for multiple actinic keratoses.

**Abstract:**

Background: Actinic keratosis (AK) is one of the most common skin diseases, with a low risk of progression into invasive squamous cell carcinoma. We aim to assess efficacy and safety of a novel formulation of 5-Fluorouracil (5-FU) 4% with once daily application for the treatment of multiple AKs. Methods: A pilot study was performed on 30 patients with a clinical and dermoscopic diagnosis of multiple AKs, enrolled between September 2021 and May 2022 at the Dermatology Departments of two Italian hospitals. Patients were treated with 5-FU 4% cream once daily for 30 consecutive days. The Actinic Keratosis Area and Severity Index (AKASI) was calculated before starting therapy, and at each follow-up, to assess objective clinical response. Results: The cohort analyzed included 14 (47%) males and 16 (53%) females (mean age: 71 ± 12 years). A significant decrease in AKASI score at both 6 and 12 weeks (*p* < 0.0001) was observed. Only three patients (10%) discontinued therapy, and 13 patients (43%) did not report any adverse reactions; no unexpected adverse events were observed. Conclusions: In the setting of topical chemotherapy and immunotherapy, the new formulation of 5-FU 4% proved to be a highly effective treatment for AKs and field cancerization.

## 1. Introduction

Actinic keratosis (AK) is one of the most common skin diseases treated by dermatologists. It is caused by chronic exposure to ultraviolet radiation, it is prevalent in fair skin types and in men over the age of 80 [1]; the majority of AKs arises on sun-exposed areas, such as extremities and head and neck area. This skin condition manifests typically as multiple lesions in a continuous area, defined field of cancerization (FC) (≥6 AKs) [1]. The clinical classification by Olsen et al. is based on the thickness of the lesions. In fact, AKs, according to this classification, are divided in grade 1 (palpable flat macules; easier felt than seen), grade 2 (red, scaly lesions, easier to see), and grade 3 (hyperkeratotic lesions) [2]. The diagnosis of AK is mainly established on clinical presentation, although dermoscopy can be a useful tool to distinguish early AKs from advanced stage [3]. In case of suspicious lesions, histopathological examination is needed to differentiate AK from squamous cell carcinoma (SCC).

The rate of neoplastic transformation of AK into invasive SCC varies between 0.025% and 16% [4]. AKs are considered a risk factor for the development of SCC, but it is difficult to predict the risk of progression of each lesion [5]. The majority of SCCs is associated with AKs and this two diseases share important similarities, such as histopathological features, cell cytology, and mutational profile [6]. Histopathological analysis have reported that clinically detectable lesions are often associated with subclinical/invisible lesions [7,8]. As supported by a recent European multicenter consensus, both single lesions and FC should be treated, taking into consideration the needs and compliance of the patient [9].

The high prevalence of AK makes the burden of the disease significant; in fact, the treatment is very important, and several therapies have been approved, such as fluorouracil cream, imiquimod cream, or photodynamic therapy. However, current guidelines do not provide standard and clear recommendations.

5-Fluorouracil (5-FU), frequently studied at different concentrations and regimens, is a pyrimidine analogue belonging to the family of antimetabolites. It acts as a pyrimidine antagonist for DNA and RNA synthesis, inducing cell apoptosis, especially in cells with high mitotic activity [10,11]. Additionally, it causes the release of cell antigens that heighten the immune system reaction, achieving an inflammatory response. This action aids in its necrotic abilities and leads to AK clearance. It is simple to apply, and it is not expensive. The 5% formulation sometimes leads to poor adherence amongst patients due to local reactions and, therefore, to treatment failure. The novel formulation of 5-FU 4% cream is a topical therapeutic option recently approved for the treatment of multiple AKs that increases patient compliance and clinical outcome [12], but, to date, real-life data on this treatment are not available. Effectiveness and tolerability of the topical treatment with 5-FU 4% in a cohort of AK patients have been investigated in a multicenter, prospective, pilot study (Protocol Code: S5FU4TAK). We also performed a review evaluating the latest developments associated with topical immunotherapy for AKs and immunological effects of topical drugs in FC.

## 2. Materials and Methods

In the period between September 2021 and May 2022, thirty patients with a diagnosis of multiple AKs were enrolled at the Dermatology Departments of two Italian hospitals, Campus Biomedico of Rome and Maggiore Hospital of Trieste, and they were treated with 5-FU 4% cream once a day for 30 days. Patients of 18 years of age or older with a clinical diagnosis of six or more AKs (grades I to II according to Olsen classification), in the head and neck area, were eligible for participation. Exclusion criteria were allergy to drug or excipients, pregnancy or breast-feeding, co-administration with brivudine, sorivudine and analogues (antivirals), DPD (dihydropyrimidinase) deficiency, and anticoagulant therapy.

The aim of the study was to assess the clinical efficacy of 5-FU 4% in the treatment of AKs within the field of cancerization at 6 and 12 weeks. The secondary outcomes were the evaluation of side effects (type and severity), adherence, and possible non-response to treatment.

The cream was applied once a day in sufficient quantities to cover the entire area affected by AKs (including face skin, ears, and scalp) with a thin layer, using the fingertip to massage the drug gently and uniformly on the skin, as written in the package leaflet.

Clinical and dermatoscopic pictures of the lesions were obtained during the first visit and after 6 and 12 weeks. We described the presence or absence of classical dermoscopic features of AK classification for each patient: perifollicular vascular pseudonetwork, white/yellowish scales, and erythematous background with enlarged follicular openings [13,14]. The Actinic Keratosis Area and Severity Index (AKASI), with a numerical score ranging from 0 (absence of AK) to 18 (AK of the severest possible degree), was calculated before starting therapy and at each follow-up (after 6 and 12 weeks) to assess objective clinical response [15].

Visual assessment and questions about patient symptoms and tolerability were used to evaluate local skin reactions for each patient. The reactions were classified at week 6 using a numerical scale: 0 for no reaction, 1 (mild) for slightly visible reaction, 2 (moderate) for discrete reaction, and 3 (severe) for intensive/worrisome/not tolerated events.

For statistical analysis, parametric or non-parametric tests were carried out, after checking the distribution of the data, as well as using skewness and kurtosis normality tests. For normally distributed variables the *t*-test or ANOVA test was used, and, for the others, the Kruskal-Wallis test was applied. Comparisons between AKASI at different times were made with the *t*-test for paired data and with the Wilcoxon test for paired data. Associations between categorical variables were tested with the Chi-square test or Fisher’s exact test, depending on the sample size (the latter in the case that one or more expected cell counts in the cross-tabulation are less than 5). *p*-values < 0.05 were considered statistically significant; statistical analysis was performed with the Stata/SE 16.0 package (StataCorp, College Station, TX 77845-4512, USA).

## 3. Results

From September 2021 to May 2022, 30 patients were enrolled: 14 males (47%) and 16 females (53%), aged between 49 and 88 years (mean age: 71 ± 12 years).

The study included eight patients (26.7%) with Fitzpatrick skin types 1, sixteen patients (53.3%) with skin type 2, and six patients (20%) with skin type 3. The majority of the patients (93.3%) had a history of previous sun chronic exposure due to occupational reasons and/or hobbies.

The clinical evaluation showed that 26 patients (86.7%) presented AKs localized exclusively on the head area, and only four patients (13.3%) had diffuse AKs on multiple sites, including the face, trunk, and limbs. Among the latter, two patients had AK on the face and limbs; the other two presented with them on the face, trunk, and limbs.

Statistical analysis revealed a significant difference in AKASI score at both 6- and 12-weeks vs. baseline (*p* < 0.0001) (Figure 1). 

The mean AKASI value at enrolment was 4.1 (range 1.8–8.4, standard deviation 2.12), at week six it was 2.1 (range 1.32–5.4, standard deviation 2.12), and, at week twelve, it was 1.4 (range 0.2–2.6, standard deviation 0.76).

Dermoscopic examination at baseline visit showed red pseudonetwork (87%) and white/yellow scales (77%) as the most frequently described criteria, followed by erythematous background with enlarged follicular openings (50%) (Table 1). At 12-week follow-up visit, 25 patients (83%) achieved complete clinical and dermoscopic clearance (Figure 2).

Treatment was generally well tolerated. Only three patients (10%) discontinued therapy due to adverse events onset (pain and burning); thirteen patients (43%) did not report any adverse reaction; sixteen subjects (53%) experienced erythema, thirteen (43%) scabs, eleven (37%) skin erosions, six (20%) bleeding, and two (7%) desquamation (Figure 3). In addition, two (7%) patients reported pain, and four (13%) reported burning.

The majority of the severe reactions at week six were related to erythema (20%), scabs (7%), and pain (7%)/burning (7%) (Table 2).

## 4. Discussion

Several therapeutic approaches for AKs have been approved in recent years. It is possible to differentiate lesion-directed therapies, including ablative procedures, and field-directed strategies, such as 5-FU, diclofenac, imiquimod, and photodynamic therapy (PDT) [5]. Immune-based interventions are one of the most effective and popular treatments for multiple AKs. The novel formulation of 5-FU 4% cream has been recently approved for the treatment of multiple AKs. 5-FU is actually the first approved drug for FC, and it is a pyrimidine analogue that inhibits thymidylate synthase; it decreases, therefore, the proliferation of atypical cells and induces apoptosis [16]. The inflammation caused by the drug increases the antitumor effect and the released cell antigens heighten the immune system reaction, achieving significant clinical responses [1]. 5-FU concentrations of 1%, 2%, and 5% are usually applied twice daily for two to four weeks, and the clearance of the lesions occurs in approximately 62.5% of patients in complete adherence to the treatment [1]. In the review of Kaur et al., the clearance rates of AKs after four weeks of administration of 5-FU 0.5% ranged from 14.9% to 57.8%, whereas the clearance rates of 5-FU 5% ranged from 43% to 100% [17]. The results of Loven et al. reported, however, severe facial irritation in 47.6% of patients treated with 0.5% cream and in 57.1% of those treated with 5% cream, and, overall, the patients preferred the 5-FU 0.5% because it was more tolerable with less irritation, but also because it was easier to apply with one daily application [18].

The 5-FU 4% cream is used to treat AK lesions (Olsen grade I-II) of the head; the indication is based on two important phase III studies (HD-FUP3B-048 and HD-FUP3B-049) [19]. The application of 5-FU 4% cream once a day resulted in comparable rates of completely (54.4% vs. 57.9%) and partially clearing AK lesions (80.5% vs. 80.2%) to the traditional 5-FU 5% cream treatment applied twice a day [12]. The side effects of these different formulations were also evaluated. At week four, severe reactions occurred more frequently with the 5-FU 5% cream applied twice a day compared to 5-FU 4% cream applied once a day [19].

The findings of the exploratory analysis of Stockfleth et al. showed that patients treated with 5-FU 5% vs. 4% cream described severe erythema (47% vs. 37%), severe scaling/dryness or crusting (both 25% vs. 18%), severe stinging/burning (27% vs. 18%), and severe pruritus (22% vs. 13%; all *p* < 0.05). Whereas, similar rates of severe swelling (8% vs. 5% of patients; *p* = 0.2081) and severe skin erosions (12% vs. 8%; *p* = 0.1495) were observed between treatment groups [19]. Additionally, it is noteworthy that the occurrence of severe local skin reactions was associated to the number of AKs at the start of the treatment [19].

In our study, the 5-FU 4% cream was associated with a clinical improvement and a reduced incidence of local reaction; in fact, 43.3% of the patients did not present any adverse effects and the others experienced common mild reactions, such as erythema, erosions, scabs, scaling, pain, burning, and bleeding. Treatment was indeed generally well tolerated; no new toxic effects were noticed. In previous studies, we observed severe skin reactions, mainly related to erythema, scabs, and pain/burning sensation. However, we observed a higher percentage of patients who did not manifest adverse events and a lower rate of severe adverse reactions compared to prior studies. Even if there are few studies related 5-FU 4%, the possible explanations for these differences may be the lower number of patients enrolled in our study and, probably, the inferior number of skin lesions at baseline [19]. It should be noted that the analysis Stockfleth et al. provides information about AK severity considering only the total number of lesions [19]. Therefore, it is difficult to compare data accurately.

Unlike the 5% concentration, in our study, we also obtained good patient compliance, probably due to the daily single-dose application and the short treatment period. Since tolerability greatly affects treatment adherence, it could be crucial in further studies to identify patients who may experience more intense skin reactions while receiving 5-FU 4% cream treatment, in order to provide appropriate information and management.

Furthermore, in our study, we assessed, for the first time, the dermatoscopic clearance of AKs after topical treatment with 5-FU 4%. Dermoscopy has become an important noninvasive diagnostic tool in clinical practice, and it plays a central role in the management of AK and NMSC, evaluating the lesion margins/severity and monitoring the effects of topical treatments. The different grades of AK severity are associated with distinct dermoscopic patterns. Grade 1 AKs are dermoscopically typified by red pseudonetwork patterns and discrete white scales, and grade 2 corresponds to an erythematous background with white-to-yellow keratotic and enlarged follicular openings. Grade 3 AK exhibits either enlarged follicular openings filled with keratotic plugs over a scaly and white-yellow background or marked hyperkeratosis seen as white yellow structureless areas [13,20]. Dermoscopy has been reported to have a high sensitivity and specificity of 98% and 95%, respectively, in diagnosing AK [21].

In our study, the treatment efficacy was highlighted by the dermoscopic reduction/disappearance of the perifollicular vascular pseudonetwork and the characteristic yellowish scales. These data support the potential role of dermoscopy in the evaluation of the effectiveness of AK treatment [22]. Previous studies have shown the importance of dermoscopy in the initial assessment and monitoring of AKs after different therapies, such as imiquimod [22], 5-FU 0.5%/salicylic acid 10% [23], PDT, or cryotherapy [24]. Up to date, our study is the first one that described, also, dermoscopic results after treatment with 5-FU 4% cream.

To objectively evaluate the baseline clinical status and the progressive improvement, we used the AKASI score. It is a system used to assess the severity of AK, based on the area and severity of AK lesions and used to monitor response to treatment. In studies investigating the efficacy of 5-FU cream for AK, the AKASI score is often used as an outcome measure [25]. It is considered a reproducible method, used as an alternative to the Total Lesion Count (TLC) [15]. The advantage is the possibility to monitor and compare disease progression in an objective manner. This is similar to established scoring systems used for other chronic diseases, such as the psoriasis area and severity index (PASI). The majority of studies and clinical trials for AKs have used either TLC or Olsen’s lesion classification. However, the main disadvantage of these systems is that they only evaluate the severity of each individual lesion and do not take into account the entire affected area of the skin [26]. Compared to other parameters, the AKASI score provides numerical value and, furthermore, it gives a numerical threshold value for skin cancer, predicting the risk of progression to SCC [26,27].

In our study, the AKASI score confirmed the progressive and significant improvement, with a reduction in the baseline score from 4.1 to 1.4 after 12 weeks.

To note, 5-FU is the only substance that has been shown to decrease the risk of SCC after therapy. Weinstock et al. indeed reported that a conventional course of 5% 5-FU, twice daily, reduced the risk of SCC for the first year after therapy [28].

Another drug commonly used as topical field-therapy is imiquimod, a toll-like-receptor-7 (TLR-7) agonist that causes secretion of cytokines and activation of the immune system [29]. This immunotherapy is associated with side effects, such as erythema, crusting, and erosion [5], similar to 5-FU. Animal and human research suggests that this drug has antiviral and antitumor properties and can increase both the innate and acquired immune responses. The production of interferon, tumor necrosis factor, interleukin-6, and interleukin-8 by imiquimod-induced cytokines can indeed activate the innate immune system [30]. Imiquimod functions as an agonist for TLR 7 and triggers proinflammatory cytokines by activating the central transcription factor, as well as nuclear factor kappa-B (NF-kB) [31]. Imiquimod interacts also with a TLR-independent signaling, including adenosine A (2A) receptor signaling pathways and reduced adenylyl cyclase activity [32,33,34].

In Italy, imiquimod cream at a concentration of 5% is approved for the treatment of anogenital warts and basal cell carcinomas, whereas the 3.75% cream has been accepted in the field-directed therapy, including sun-exposed areas, such as the face and balding scalp [35].

In 2019, Jansen et al. evaluated the efficacy of 5-FU at 5% comparing it with imiquimod cream at 5%, as well as photodynamic therapy with methyl aminolevulinate and ingenol mebutate at 0.015%. The study showed that 5% 5-FU was significantly more effective than the remaining therapies in the treatment of multiple AKs in a continuous area [11].

An emergent immunotherapeutic option for AKs is the combination of calcipotriol with 5-FU. By inducing the expression of thymic stromal lymphopoietin (TSLP) cytokine in keratinocytes, calcipotriol synergizes with the cytotoxic activity of 5-FU and leads to a strong CD4+ T cell immunity against AKs [36]. Erythema induced by calcipotriol plus 5-FU therapy is one of the most common expression of immune response against atypical keratinocytes [37]. The findings of Rosenberg et al. and Cunningham et al. demonstrated the efficacy of a short course (four-day course) of this field-therapy with mild adverse events; in fact, the 0.005% calcipotriol ointment with 5% 5-FU cream twice daily for four days induced a tissue-resident memory T cell response with a decrease in risk of SCC development on the face and scalp within three years after treatment [36,37], but further studies are needed to establish the efficacy of this immunotherapy compared with current treatment options.

## 5. Conclusions

AK is a common skin condition caused by cumulative exposure to UV radiation usually affecting old patients. Several treatments are used in order to manage this pathological lesion. All treatments are effective and well tolerated, although there are notable variations in clearance rates, tolerability, and types and frequency of side effects.

As elderly people with chronic diseases and many morbidities represent a significant percentage of the population, it is important to find interventions to improve medication adherence. Management decisions are often customized to the individual. In the setting of topical chemotherapy and immunotherapy, the new formulation of 5-FU 4% represents a highly effective treatment for AKs and field of cancerization, achieving good patient compliance and significant treatment outcome. In our study, the AKASI score confirmed progressive and significant improvement. The treatment efficacy was highlighted by the dermoscopic reduction/disappearance of the perifollicular vascular pseudonetwork and of the characteristic yellowish scales.

Even if the progression of individual AK to invasive SCC is not predictable, the treatment of the lesions may be a strategy for the prevention of invasive cancer.

## Figures and Tables

**Figure 1 cancers-15-02956-f001:**
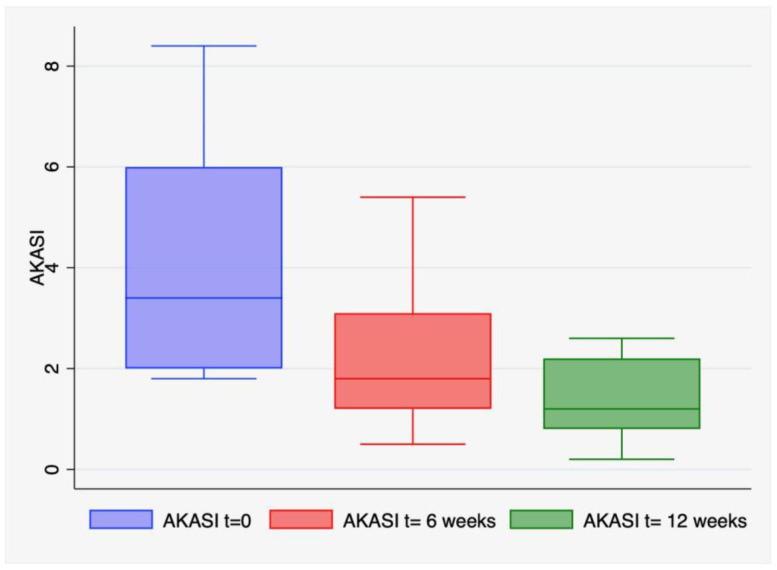
Box-plot representing the Actinic Keratosis Area and Severity Index (AKASI) score at different time points, namely, before and after six and twelve weeks of treatment. A one-way repeated measures ANOVA was run on a sample of 30 patients to determine if there were differences in AKASI score due to treatment with 5-FU 4% cream. The results showed that the treatment elicited statistically significant differences in mean AKASI score over its time course, F (2, 53) = 35.7, *p* < 0.0001.

**Figure 2 cancers-15-02956-f002:**
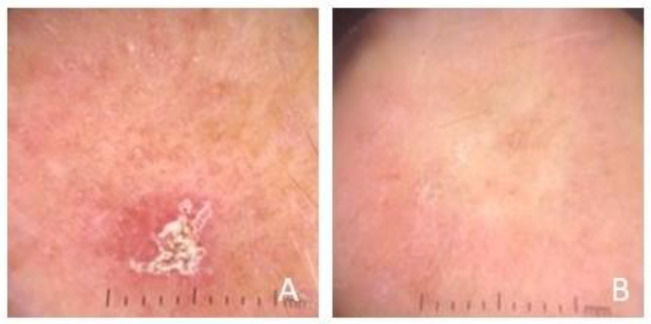
Example of complete dermoscopic clearance at week 12. (**A**) Baseline dermoscopic features of AK: erythematous background with yellowish scales. (**B**) Dermoscopic disappearance of AK features at week 12.

**Figure 3 cancers-15-02956-f003:**
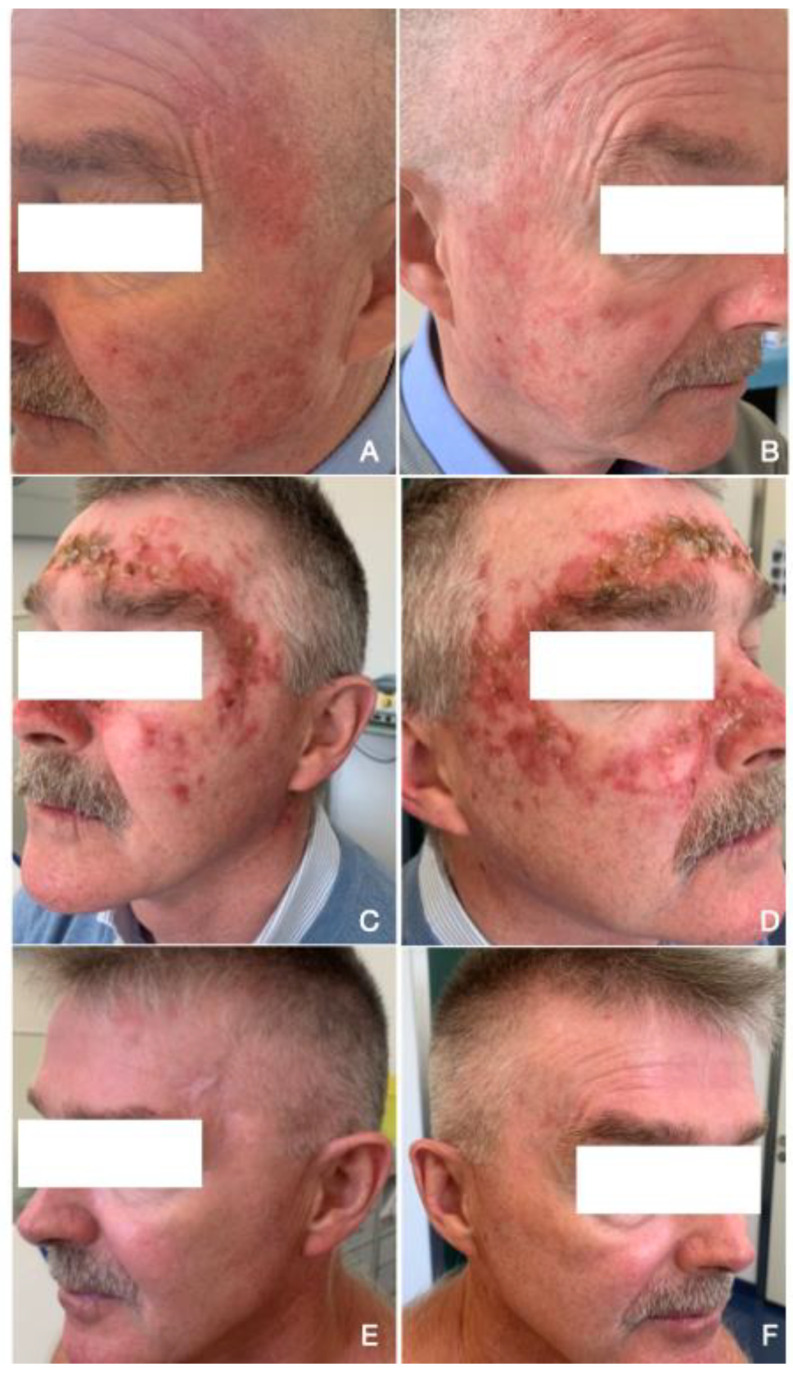
Example of patient with more than six actinic keratoses on the face treated with 4% 5-Fluorouracil cream. (**A**,**B**) Clinical presentation of multiple AKs before treatment. (**C**,**D**) Common side effects with 5-FU 4% cream: erosion, erythema and scabs. (**E**,**F**) Complete response after six weeks of treatment.

**Table 1 cancers-15-02956-t001:** Dermatoscopic features described at baseline, at week six and week twelve.

	Perifollicular Vascular Pseudonetwork (%)	White/Yellowish Scales (%)	Erythematous Background/Enlarged Follicular Openings (%)
**Baseline**	26 (87)	23 (77)	15 (50)
**Week 6**	10 (33)	9 (30)	7 (23)
**Week 12**	3 (10)	2 (7)	2 (7)

**Table 2 cancers-15-02956-t002:** Local reactions at week six of treatment with 5-fluorouracil 4% cream.

Grade of Reactions *	Erythema (%)	Scabs (%)	Erosion (%)	Bleeding (%)	Scales (%)	Pain (%)	Burning (%)
Severe	6 (20)	2 (7)	2 (7)	1 (3)	1 (3)	2 (7)	2 (7)
Moderate	4 (13)	4 (13)	5 (17)	1 (3)	1 (3)	0 (0)	1 (3)
Mild	6 (20)	7 (23)	4 (13)	4 (13)	0 (0)	0 (0)	1 (3)
None	14 (47)	17 (57)	19 (63)	24 (80)	28 (93)	28 (93)	26 (87)
Total	16 (53)	13 (43)	11 (37)	6 (20)	2 (7)	2 (7)	4 (13)

* The local reactions were classified at week six using a numerical scale: 0 for no reaction, 1 (mild) for slightly visible reaction, 2 (moderate) for discrete reaction, and 3 (severe) for intensive/worrisome/not tolerated events.

## Data Availability

The data presented in this study are available upon request from the corresponding author.

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
