# Peer review of "Actinic Keratoses: A Prospective Pilot Study on a Novel Formulation of 4% 5-Fluorouracil Cream and a Review of Other Current Topical Treatment Options"

_cancers, 2023, doi:10.3390/cancers15112956_

Round 1

Reviewer 1 Report

The authors present a case series of 30 patients with actinic keratoses who have been treated with topical 4% 5-FU for 4 weeks and monitored week 6 and week 12.

Although the number of patients is low and the follow up time is short, I think there is some benefit to this study in case of an intensive revision.

1) I am surprised that only few data are shown, more of the potentially interesting data need to be presented:

1.1) dermatoscopic results: they are only mentioned in a text paragraph. A tabulated summary of the data and a representative patient should be shown.

1.2) other efficacy parameters than the AKASI score should be shown, e.g., clearance rate.

1.3) The treatment area in the patient demonstrated in Figure 1 is very large. Information on the treatment areas in all patients and correlation with efficacy/adverse events is important.

1.4) The adverse events are summarized in one paragraph without grading. This should also be given in a table.

2) The discussion is very long and rather a lecture on therapeutic options in actinic keratoses than the discussion of the results.

2.1) The discussion can be focused and shortened.

2.2) Please also include a discussion on endpoints in AK-trials. E.g., what is the relevance of the AKASI-score in comparison to other parameters used in other studies?

2.3) Moreover, the role of dermatoscopy in the diagnosis and follow up of patients with AK can be discussed on the basis of the data from this trial.

3) The language should be improved. Only few examples from the introduction:

3.1) lin 39: "skin" is double

3.2) line 43: full stop after "lesions"

3.3) line 54: "this two disease"

Reviewer 2 Report

The authors present data about treatment of AK with 4% 5-FU, a novel formulation, recently approved for AK treatment (2020 in Germany). Up to now there were only a few papers published, concerning efficacy and tolerability of the drug. In this study on 30 patients treated once daily for 30 days high clinical efficacy was reported, as indicated by AKASI scores, that dropped from 4.1 to 2.1 after 6 weeks and to 1.4 after 12 weeks, reduction of perifollicular vascular pseudonetworks, and complete disappearance of yellowish scales. The authors also reported the treatment was well tolerated, resulting in good compliance of the patients. Only 3 patients (10%) discontinued treatment and more than 40% did not report any adverse reaction.

Major comments

Results: Although only a few clinical studies on 4% FU creme were yet published, the author should compare their findings with results from another 2 studies: Dohil MA (2016, J Drugs Dermatol. 2016 Oct 1;15(10):1218-1224.) and Stockfleth E et al. (2022, Dermatol Ther (Heidelb). 2022 Feb;12(2):467-479. doi: 10.1007/s13555-021-00668-9.). Results of the present study were presented in a pretty short form. In order to compare them with other studies, data about patients with complete and partial clearance should be given, as well as more detailed information about safety and side effects (meaning the number of patients reporting a specific reaction by severity, like mild, moderate, severe or not at all).

Discussion: In the 1st part of the discussion clearance rates and side effects of different 5 FU concentrations (0,5% and 5%) reported in published studies are presented and compared with results of the present study. The authors, however, should also discuss their findings on efficacy and local skin reactions regarding results from studies using 4% FU (see above). For instance, what reasons might account for 43% of patients with no reported side effects vs <10% in Stockfleth et al. 

Discussion: In the 2nd part of the discussion, some other drugs commonly used for topical treatment of AK were described. Although the section contains useful information, it does not refer to the present study on 4% FU. What I would suggest is to discuss insights from a comparison of 4% FU with other topical treatments with respect to application, safety, efficacy, treatment duration and costs.

Discussion: The paragraph on Resiquimod, which is not approved for AK, may be omitted as it is not relevant for the discussion of clinical application of 4% FU.

Discussion: The following parapraph on HPV and the putative role in KC development is also not relevant for the discussion of 4% FU study. Instead, it contains some statements that would require clarification. The predominant HPV types in AK and cSCC belong to the genus beta HPV (e.g. see Howley PM & Pfister HJ Beta genus papillomaviruses and skin cancer. Virology 479–480, 290–296, doi:10.1016/j.virol.2015.02.004 (2015). In most studies on NMSC, mucosal HPVs, including HPV 16 and 18, were rarely detected (with the exection of periungual and digital cSCC). Studies cited by the authors as a reference for detection of HPV 16 and 18 DNA in KCs, should be interpreted with caution, because FFPE samples were analyzed and nested PCR was performed, both known to be associated with a risk of contamination.

I would suggest to omit the HPV section also, as it is not necessary for the discussion of topical agents to treat AK. As the role of HPV in KC is still not clear, the discussion of HPV as a starting point for prevention and treatment of KC needs a much more comprehensive consideration of studies on epidemiology and oncogenicity of beta HPV.

Minor comments

Orthograpical errors in line 40 (“skin“ twice), line 54 (“diseases“), line 150 (“%“ twice), line 179 (“also“)

Discussion lines 175-177: Imiquimod is a ligand for TLR 7 only, but not for TLR 8 (see: Lee J. et al., 2003. Molecular basis for the immunostimulatory activity of guanine nucleoside analogs: Activation of Toll-like receptor 7. PNAS, 100(11):6646-51).

Reviewer 3 Report

The paper presents a clinical study on patience with actinic keratosis.

The topical application of 5-FU for cancer treatment is known. Maybe on the skin problems such as AK is not new approach. However this is the original study it has valuable results. 

The paper looks very short for the article. My only recommendation is to write more discussion and to presents also other images with the patients. 

This paper can be move to section "Clinical cases".

Author Response

please, see the attachment

Round 2

Reviewer 1 Report

The manuscript has been markedly improved. However, there are still some issues that need improvement:

1) The clearance rates (partial/complete) in the 30 patients should be given, as well as the lesion counts before treatment and at week 6 and 12.

2) The treatment areas in the 30 patients should be given.

3) Figure 1: what is the meaning of the parameters in the box plot (e.g., median, ICR, SEM, SD, range)?

4) The discussion can be focused on 5-FU, the information from line 281-343 can be omitted.

Round 3

Reviewer 1 Report

Thank you for the explanations. I still see some need for language improvement. E.g., see the following to sentences from the conclusion where I crossed words that do not fit:

"In the setting 317 of topical chemotherapy and immunotherapy the new formulation of 5-FU 4% represents a highly effective treatment for AKs and field of cancerization, achieving good patient compliance and significant treatment outcome. In our study The AKASI score confirmed the progressive and significant improvement."